# DNA Damage and Its Role in Cancer Therapeutics

**DOI:** 10.3390/ijms24054741

**Published:** 2023-03-01

**Authors:** Jaeyoung Moon, Ichiwa Kitty, Kusuma Renata, Sisi Qin, Fei Zhao, Wootae Kim

**Affiliations:** 1Department of Integrated Biomedical Science, Soonchunhyang Institute of Medi-bio Science (SIMS), Soonchunhyang University, Cheonan 31151, Chungcheongnam-do, Republic of Korea; 2Magister of Biotechnology, Atma Jaya Catholic University of Indonesia, Jakarta 12930, Indonesia; 3Department of Pathology, University of Chicago, Chicago, IL 60637, USA; 4College of Biology, Hunan University, Changsha 410082, China

**Keywords:** DNA damage, cancer therapeutics, mutations

## Abstract

DNA damage is a double-edged sword in cancer cells. On the one hand, DNA damage exacerbates gene mutation frequency and cancer risk. Mutations in key DNA repair genes, such as breast cancer 1 (*BRCA1*) and/or breast cancer 2 (*BRCA2*), induce genomic instability and promote tumorigenesis. On the other hand, the induction of DNA damage using chemical reagents or radiation kills cancer cells effectively. Cancer-burdening mutations in key DNA repair-related genes imply relatively high sensitivity to chemotherapy or radiotherapy because of reduced DNA repair efficiency. Therefore, designing specific inhibitors targeting key enzymes in the DNA repair pathway is an effective way to induce synthetic lethality with chemotherapy or radiotherapy in cancer therapeutics. This study reviews the general pathways involved in DNA repair in cancer cells and the potential proteins that could be targeted for cancer therapeutics.

## 1. Introduction

DNA, as genetic material, plays a major role in living organisms and needs to be maintained to transmit hereditary information. However, DNA is prone to damage that occurs either endogenously or exogenously. There are several types of DNA damage, including single-strand DNA breaks (SSBs), double-strand DNA breaks (DSBs), DNA-protein crosslink (DPC), bulky adducts, and base mismatch [1,2]. DNA damage changes the sequence of DNA, leading to the disruption of proteins and their functions [3]. Accumulation of DNA damage can have several harmful effects on human health, resulting in senescence, aging, apoptosis, and genomic instability [4,5]. The accumulation of this phenomenon leads to severe cancer progression in normal cells [6]. However, the DNA damage that occurs in cancer cells has remarkable connections with cancer patients. If the cancer cell cannot restore and activate the DNA damage response (DDR), the affected cancer cell dies [7,8]. 

The activation of DDR promotes cell repair, by which numerous proteins that play a role in this pathway assemble in the same region where damage occurs. These proteins perform a repair mechanism based on the type of damage (Figure 1) [2,9]. SSBs can be directly or indirectly repaired by base excision repair (BER) [9,10]. DSBs can be repaired using two types of mechanisms: error-prone or non-homologous end-joining (NHEJ), and less error-prone or homologous recombination (HR) [9]. Other damage types, such as DNA adducts, crosslinks, and oxidized bases, can be repaired using nucleotide excision repair (NER). When a DNA mutation such as an insertion, deletion, or base mismatch occurs, mismatch repair (MMR) is activated [2,9]. These mechanisms can be used by cancer cells to repair the damage caused by drugs or other therapies [2].

Cancer is a severe illness with high rates of incidence and mortality, thus necessitating research on the development of cancer treatments. This has led to the development of four main cancer treatments: surgery, chemotherapy, radiotherapy, and immunotherapy. Since their discovery in the 19th and 20th centuries, surgery and radiotherapy have been used to treat cancer, but their success rates are low. Since the discovery of chemotherapy, an increase in the number of cancer survivors has been recorded [11,12]. 

However, this form of treatment may have numerous negative effects on patients. In addition to damaging cancer cells, it can harm other cells, thereby causing toxicity and loss of function in healthy cells and tissues. Therefore, more research on precise and targeted cancer therapies that are safe for normal cells and have minimal side effects is required [11,12]. These targeted cancer therapies need to have higher efficacy rates than previous treatments by targeting the tumorigenic pathway to inhibit cancer growth and eliminate the cancer [13]. Certain tumorigenic pathways can be targeted for this precision therapy, including the DNA damage pathway. Understanding the mechanism of the cancer damage pathway can aid in the development of treatments specific to cancer cells [14].

## 2. Main DNA Repair Pathways in Cancer Cells

### 2.1. Homologous Recombination

HR is a mechanism for recovering DNA double helices without the loss of genetic information. The HR is recovered using the genetic information of the cloned DNA or homologous chromosomes, and occurs during DNA replication. Therefore, it occurs mostly in the S and G2 phases of the cell cycle [15]. Owing to challenges associated with precise chromosome segregation during cell division, DSBs are the most genotoxic type of DNA lesion. Although this is a relatively accurate and effective repair process, sister chromatid DNA must be present for HR repair to take place [16]. If DSBs occur, 53BP1 is activated, and the 53BP1-RIF1-shieldin-CST/ASTE1 complex stabilizes chromatin at the DSB site [17,18,19]. The MRE11-RAD50-NBN (MRN) complex detects DSBs first, recruiting the BLM helicase and EXO1 onto the breaks to initiate 5′–3′ double-stranded DNA resection. The overhang of 3′ single-stranded DNA (ssDNA) was covered by RPA, thereby preventing additional resection. Through the RPA-ATRIP interaction, ATR localizes to RPA in damage sites and activates the ATR-Chk1 DNA damage checkpoint. This causes cell cycle arrest and preserves the cell strand [20]. RAD51 recruitment is mediated by BRCA1, BRCA2, BARD1, PALB2, and RAD51, and override RPA from the 3′ overhangs to form presynaptic filaments. Subsequently, strand invasion occurs and causes the development of a D-loop between the homologous chromosome and the invading 3′ overhang strand [21]. Then, DNA synthesis and ligation occur on each of the resected ends using the template. The BTRR dissolvasome disintegrates the Holliday junctions connecting sister chromatids and completes HR repair [20]. HR deficiencies are more susceptible to destruction by DNA-damaging agents or by substances that block other repair pathways or checkpoint systems [22,23]. HR needs several mediator proteins, such as BRCA1 and BRCA2. Multiple cancers, including breast, ovarian, and pancreatic cancer, involve altered HR genes [24]. Therefore, HR proteins are prospective targets for cancer therapeutics due to their roles in tumorigenesis and their involvement in therapeutic resistance [25].

### 2.2. Non-Homologous End Joining

DSBs can be restored through a mechanism called NHEJ. NHEJ pathways in the DDR pathway are less accurate, but still effective, and can introduce DNA rearrangements. In addition, they did not require duplicated DNA [26]. To maintain genomic stability, DSBs must be repaired quickly; therefore, cells use the NHEJ pathway to repair DSBs. However, NHEJ contains errors, because several bases can be inserted or deleted. A homologous template is not required because the break ends are directly ligated. Although NHEJ is involved in the cell cycle, the G1 phase is more significant. The first step of NHEJ is the recognition of DSB by Ku, a heterodimer comprising Ku70 and Ku80 proteins. Ku interacts with the DNA ligase IV complex and XRCC4-like factor (XLF) and may serve as a docking site for another NHEJ protein. XLF is involved in NHEJ ligation steps, interacting with DNA ligase IV and XRCC [27]. When Ku is recruited to a DSB, DNA-PKcs are attached to create the DNA-PK complex, which undergoes autophosphorylation, and phosphorylates NHEJ factors to draw them to the DSB for repair. In the NHEJ environment, XRCC4’s functions also make it easier to recruit components for the DSB. When LIG4 and XRCC4 are bound together, LIG4 ligase activity increases, sealing the blunt or homologous overhang ends of the DSB. XRCC4-LIG4 interacts with XLF and PAXX [28]. The Artemis DNA PKcs complex, which has various endonucleolytic properties, functions as a nuclease [29]. The polymerization that takes place throughout NHEJ is accomplished by the Pol X family of polymerases, and Pol l is preferred because it can function in a template-independent manner. The DNA ligase IV-XRCC4-XLF complex is responsible for the final closure of the DNA break [30]. 

NHEJ stabilizes the genome in normal cells, but promotes genomic instability and carcinogenesis in cancer cells. Elevated Ku expression increases tumor proliferation and metastasis and results in shorter survival duration [31]. For instance, in non-melanoma skin cancer, up-regulation of Ku70 and Ku80 protein levels is correlated with the tumor proliferation rate. Meanwhile, mutations in genes that participate in NHEJ lead to hereditary breast and ovarian cancers. Cancer progression and low survival rates have been linked to increased expression of DNA-PKcs. DNA-PKcs expression at the mRNA level is much higher in NSCLC tumor tissues than in the surrounding normal tissues, and the increased expression is linked to a higher mortality risk [32]. Therapies aimed at the NHEJ pathway may be used to target tumor cells that depend on this pathway [33].

### 2.3. Microhomology-Mediated End Joining

MMEJ is an alternative non-homologous end joining (Alt-NHEJ) mechanism that is used to repair DSBs. When the broken ends are aligned before joining MMEJ, microhomologous sequences are used, resulting in deletions on either side of the initial break. Chromosome anomalies such as translocations, deletions, inversions, and other intricate rearrangements are typically linked to MMEJ. The utilization of 5–25 base pair (bp) microhomologous sequences during the alignment of broken ends before joining, resulting in deletions flanking the initial break, is the primary characteristic that sets MMEJ apart. Chromosome abnormalities such as deletions, translocations, inversions, and other intricate rearrangements are frequently linked to MMEJ [34,35,36,37,38,39]. In MMEJ, end resection by the MRE nuclease, which leaves single-stranded overhangs, initiates the repair of the DSB [40]. Microhomologies, which are brief areas of complementarity (often 5–25 base pairs) between the two strands, are where these single-stranded overhangs anneal. Polymerase theta-mediated end-joining (TMEJ), a type of MMEJ, can repair breaks [41,42]. The DNA polymerase theta helicase domain has single-strand annealing activity that is ATP-dependent and may encourage the annealing of microhomologies [43]. Overhanging bases are eliminated by nucleases such as Fen1 after annealing, and gaps are filled by DNA polymerase theta [44]. Polymerase theta’s capacity to fill gaps contributes to the stabilization of the annealing of ends with little complementarity. In addition to microhomology footprints, the mutational signature of polymerase theta includes templated inserts, which are believed to be the result of a template-dependent extension that failed and was then re-annealed at secondary homologous sequences [42]. 

MMEJ is an inherently mutagenic mechanism for DSB repair. In primary human cancer cells, oncogenic chromosomal translocation breakpoints carry microhomology signatures, suggesting that MMEJ may be the mechanism causing this translocation [45,46]. DNA polymerase θ (Polθ) is a protein that promotes MMEJ. Polθ overexpression in breast, lung, bladder, colorectal, gastric, glioma, pancreatic, prostate, melanoma, and uterine malignancies is associated with poor prognosis. As NHEJ-deficient cells also rely on MMEJ, the application of a Polθ inhibitor is possible for the treatment of cancer. Collectively, these findings offer a compelling reason to focus on MMEJ for malignancy therapy, especially for tumors resistant to PARP inhibitors [22,47].

### 2.4. Base Excision Repair

BER is the most adaptable excision repair mechanism and oversees the fixation of most endogenous lesions, including oxidized bases, AP sites, and DNA SSBs. The fundamental BER mechanism in E. coli was discovered for the first time [48]. The BER pathway is responsible for repairing DNA SSBs, which are most frequently caused by modified bases, abasic sites, and their processing of more than 20,000 events per cell per day [49,50]. BER is a type of repair that simply removes the base, and is distinguished into two techniques: short and lengthy patch repair. Long patch repair fixes 2–10 nucleotides, whereas short patch repair fixes only one. The primary path is typically a brief patch fix, and simply performs the following actions: excision, incision, end processing, repair synthesis, and base lesion. DNA glycosylase, AP endonuclease, DNA polymerase, and DNA ligase are key enzymes. First, aberrant bases, such as uracil bases, are identified and cleaved by DNA glycosylation enzymes [51]; an apyrimidine (AP) is present. Cleavage of the damaged N-glycosidic bond of the base results in the creation of an AP site. AP spots can be recognized by AP endonucleases such as APE1 [52]. Therefore, the phosphodiester bond was broken. APE cleaves to the AP site to generate 3′-OH and 5′-deoxyribose phosphate (dRP) termini. The intrinsic dRP lyase activity of DNA polymerase β (Pol β) cleaves the dRP residue to produce 5′-phosphates. DNA ligase I is more crucial than DNA ligase III for BER [53,54]. Therefore, the 5′ to 3′ exonuclease activity of DNA synthase I can repair damaged bases and restore them to normal bases. DNA ligase ligates the nick; however, this repair may result in structural distortions or ground-level issues.

The BER pathway repairs residues damaged by ROS, IR, and alkylating agents. ROS-induced DNA damage is thought to play a role in the development of cancer, aging, and neurodegeneration. DNA oxidative stress can result in mutations that turn tumor suppressor genes on or off [55]. The likelihood of genetic changes resulting in neoplastic events is influenced by numerous DNA repair mechanisms as well as other cellular stress response pathways, including cell cycle arrest and apoptosis. Numerous types of DNA damage have been firmly linked to tumor development. Therefore, BER could be of critical importance for cancer prevention [56]. 

Typically, the intermediates of the BER pathway are more hazardous than the original base lesion. As a result, altering the amounts of BER proteins may be a viable gene therapy strategy for eliminating cancer cells. Other potential pharmacological targets for cancer therapy are Pol β, APE1, and DNA ligases [57,58,59]. Analysis of BER gene mutations in cancers might be useful to comprehend the genesis of malignancies in a specific organ and, more importantly, the potential function of BER in metastasis. Additionally, BER enzymes are crucial targets for cancer drugs, as they prevent cell sensitivity to several chemical substances and ionizing radiation (IR).

### 2.5. Nucleotide Excision Repair

The NER pathway, which predominantly addresses UV-induced damage, also plays a significant role in addressing DNA damage caused by platinum salts, and deals with mutated nucleotides that alter the structural integrity of the double helix [60]. UV causes DNA damage. This effect leads to the formation of thymine dimers. Thymine dimers cause severe distortion of the DNA strand. NER is a mechanism for removing this type of damage. UvrAB moves along the DNA and identifies distortions. When a damaged area is encountered, UvrA is released, and UvrC is combined to form UvrBC. UvrBC breaks the positions of 4′–5′ nucleotides to 3′ and 8′ nucleotides to 5′ at the thymine dimer site to create a gap. Helicase activity in UvrD eliminates damage and releases UvrB and UvrC. DNA polymerase I binds to repair the gap, and DNA ligase connects the nick. The thymine dimer is then repaired. 

NER may remove a variety of helix-distorting DNA lesions that are mostly caused by environmental mutagens such as ultraviolet light (UV) irradiation and large chemical compounds [61]. If UV irradiation is not controlled, in addition to causing regular cell death, it can disrupt DNA integrity and cell and tissue homeostasis, resulting in oncogene and tumor-suppressor gene mutations. If left unchecked, these mutations can result in aberrant cell proliferation and increase the probability of cancer development [62,63]. According to extensive tumor mutation research, NER may underlie a variety of mutational signatures [64,65]. Genetic variation or mutations in nucleotide excision repair genes can influence cancer risk. Therefore, cancer susceptibility may result from hereditary polymorphism changes in NER genes [66].

### 2.6. Mismatch Repair

A technique called MMR is used to identify and correct base errors that may occur during the replication and recombination of DNA, as well as various types of DNA damage [67,68]. The MMR pathway addresses replication mistakes, such as nucleotide insertions and deletions, as well as mismatched base pairing [69]. First, MutS determines the base pair error. The MutS/MutL complex binds to MutH, which is already bound to a semi-methylated sequence. MutH is then activated to cleave the unmethylated strands. The exonuclease removes the nascent strand from the cut point to the error base-pair portion, and DNA polymerase III fills the resulting gap. The DNA ligase then connects the nick, and the error base pair is repaired normally. The fact that reduction in the MMR protein expression causes a predisposition to colorectal, gastric, endometrial, and ovarian malignancies emphasizes the crucial function of MMR in carcinogenesis. Furthermore, 15% of all primary tumors contain MMR deficiencies [70]. Base substitution and frameshift mutations are greatly elevated in the mutator phenotype, which is caused by microsatellite instability (MSI) and mismatch repair deficiency. Microsatellites, which are found throughout the genome, are brief tandem repeating DNA sequences of 1–4 base nucleotides. These repetitions have a high error rate during replication, and if tumor suppressor genes contain them, a poor repair could have negative consequences [71]. 

### 2.7. DNA-Protein Crosslinks

DPCs are formed when proteins covalently bond to DNA strands. Crosslinks are particularly dangerous because they can successfully stop DNA replication and gene transcription. IR, UV rays, and other transition metal ions, such as chromium and nickel, can generate DPCs [72]. Furthermore, DPCs are frequently created by interactions with aldehydes and binding of different enzymatic intermediates to DNA, and can cause severe mutations and cell death if not repaired promptly [73]. DPC repair involves HR and nucleotide excision. Ruijs-Aalfs syndrome and Fanconi anemia are related to deficiencies in DPCs repair pathways. 

The amines of DNA bases can react with acetaldehyde, an important metabolite of ethanol and an intermediate in glucose metabolism, to produce DPCs [74,75,76]. The repair or avoidance of DNA adducts created by acetaldehyde has been found to depend on the NER, HR, and Fanconi anemia (FA) pathways [74]. Multiple chromosomal instability disorders, an increase in bone marrow loss, and a propensity for malignancy are the hallmarks of FA [77,78,79]. Several genotoxic effects, including chromatid breaks and chromosomal abnormalities and mutations, result from DPCs that are not repaired [80]. DPCs need various FA proteins to complete the repair [81].

## 3. DNA Repair Pathways Are the Achilles Heel of Cancer under Treatment

DNA lesions occur in several forms, including insertion or deletion mismatches, SSBs, and DSBs [82]. DNA lesions that cannot immediately be repaired could generate miscellaneous mutations. These mutations can cause genomic instability, which is the primary driver of cancer development and progression [83]. Considering that every cell is easily exposed to various carcinogens, both endogenously and exogenously, cells have developed numerous DNA repair pathways, referred to as DDR, that allow for their survival [84,85]. Contingent upon persistent DNA damage, normal cells undergo either apoptosis or senescence as an outcome of DDR (Figure 2). The lack of proper DDR after exposure to stressors may result in elevated occurrence of genomic instability and mutations, further injury to the DNA repair ability, and escalation of cancer development [86]. Mutated DNA repair genes are frequently detected in human cancers (Table 1), indicating that the dysregulation of DNA repair factors promotes cancer progression [87].

Even though mutations in the DNA repair system could lead to the development of certain cancers, these mutations could be a weakness for cancer cells as well. Multiple cancer cells with DDR alterations have been shown to be more sensitive to genotoxic stress generated from immunotherapy, radiotherapy, and/or chemotherapy [88]. For instance, mutations in *BRCA1* and/or *BRCA2* increase the risk of breast and other cancers, such as ovarian and prostate cancer [89,90]. Patients with cancer that have alterations in the *BRCA1* and/or *BRCA2* genes are highly sensitive to platinum chemotherapy and PARP inhibitors [91,92]. BRCA2’s mutation partner and localizer (PALB2) is associated with pancreatic and breast cancer malignancies. Cancer patients with mutations in these genes tend to receive more benefits from chemotherapeutic approaches, including platinum-based chemotherapy (NCT 03140670), mitomycin C, and cisplatin [93,94]. Cancer cells with the BRCAness phenotype do not have BRCA germline mutations, but share a similar phenotype with BRCA germline mutations and, as a consequence, this type of cancer exhibits defective HR [95]. For example, cancers with *ATM*, *ATR*, and *TP53* mutation; *METTL16* overexpression; *PTEN* deletion; and *RAD51C* hypermethylation could lead to the forfeit of the HR repairing system [96,97]. Moreover, tumors with the BRCAness phenotype are sensitive to DNA-damaging agents such as cisplatin, mitomycin C, and PARP inhibitors [98].

KU70 and KU80 mutations are associated with higher genomic instability and eventually facilitate the development of cancer. *KU70* and *KU80* polymorphisms are found in several types of cancer, such as breast, prostate, oral, bladder, colon, and lung cancers [99,100,101]. Cancer cells with mutations in either *KU70* or *KU80* are found to be more sensitive to IR [102]. Mutations on the tumor suppressor gene *ATM* are associated with a broad range of human cancers, such as lung, colorectal, hematopoietic, and breast cancers. Patients with an *ATM* loss of function are hypersensitive to IR [103]. *ATR* mutations have been detected in endometrial cancer, and cancer cells with defective *ATR* are vulnerable to several DNA-damaging chemotherapy agents and IR [103,104]. Meanwhile, DNA mismatch repair deficiency (MMRd) is also associated with several types of cancer, such as hereditary nonpolyposis colorectal cancer (HNPCC) and colorectal cancer (CRC) [105]. Any mutation on *MLH1*, *MSH2*, *MSH6*, or *PMS2* that could generate an MMRd tumor is sensitive to immunotherapy with checkpoint inhibitors [106]. Furthermore, DNA polymerase epsilon (Pol ε) and MutY DNA glycosylase (MUTYH) are involved in a BER repair system [107,108]. The mutated *POLE* gene, which encodes Pol ε, is known to initiate a hypermutator phenotype in cancers such as endometrial cancer [109,110]. Mutations in this gene are sensitive to immune checkpoint inhibitors (ICIs) [111]. Meanwhile, mutations in *MUTYH* could damage its glycosylase activity and diminish its capacity to eradicate mispaired bases, which increases the risk of cancers such as pancreatic ductal adenocarcinoma (PDAC) and CRC [108,112,113]. Tumors with *MUTYH* mutations may efficiently respond to ICI treatment [114]. Furthermore, the *ERCC2* mutation could lead to the loss of cellular NER capacity and bladder cancer development [115]. Patients with bladder cancer who have somatic *ERCC2* mutations have a higher sensitivity to cisplatin-based neoadjuvant chemotherapy [116]. 

**Table 1 ijms-24-04741-t001:** DNA damage response alterations in several cancers.

DDR Pathways	DDR Genes	Type of Alterations	Cancer Type	Reference
HR	*BRCA1*	c.190T>C (cysteine to arginine)	Breast cancer	Wang et al. [117]
	*BRCA2*	c.6408delA (deletion)		
	*BRCA1*	c.4837A>G (serine to glycine)	Ovarian cancer	
	*BRCA1*	c.2612C>T (proline to leucine)		
	*BRCA2*	c.677delC (deletion)	Esophagus cancer	
NHEJ	*Ku80*	Ku80 polymorphism G-1401T (SNP rs828907)	Oral, colon, gastric, breast, and bladder cancer	Sischc and Davis [100]
	*Ku70*	Ku70 C-61G polymorphism (SNP rs2267437)	Breast, prostate, and lung cancer	
BER	*POLE*	Missense mutation in *POLE* exonuclease domain	Endometrial cancer	Leon-Castillo et al. [110]
	*MUTYH*	G: C to T: A transversions mutation		Villy et al. [118]
NER	*ERCC2*	Somatic *ERCC2* missense mutation	Muscle-invasive bladder cancer	Mouw [115]
MMR	*MLH1*	Loss of MLH1, MSH2, and MSH6 due to *MLH1* promoter hypermethylation	Ampullary and colon cancer	Ruemmele et al. [119]

For several years, radiotherapy and chemotherapy have been used to eliminate or at least reduce the number of cancer cells; however, these methods have several drawbacks. Many tumor types remain insensitive to both methods, which causes the success rate to differ depending on tumor type and grade [120]. Naturally, cells have several DDR pathways that can moderately compensate for each other [121]. Hypothetically, cancer cells already exhibit DDR, which contributes to genomic instability. The deficiencies of a single DDR could be resolved by the compensatory pathway, making cancer cells overdependent on that pathway [122]. The concept of synthetic lethality refers to a situation where two or more genes are mutated, and cell death occurs only when both genes are mutated simultaneously. For example, there are two crucial DNA repair pathways to repair DSB HR and NHEJ. DSBs can be lethal for the cell when both DNA repair systems are inhibited and cell death is then triggered [123]. Thus, synthetic lethality induction could be exploited as an alternative to overcome the limitations of chemo- or radiotherapy by targeting the compensatory pathways, which would prevent cells from repairing and elevate cancer cell vulnerability to radio- and/or chemotherapy, leading to the apoptosis of the cancer cells [23,124,125]. 

## 4. Developing Inhibitors Targeting Key Enzymes in DNA Repair Pathways for Cancer Therapeutics

Cancer cells have several strategies that allow them to develop some ability to withstand cancer treatment. As previously mentioned, the induction of synthetic lethality in cancer cells could improve the effectiveness of cancer treatments. In line with the synthetic lethality concept, DDR inhibitors have been developed to target specific genes through specific mechanisms that block compensatory DNA repair pathways and subsequently induce the death of cancerous cells (Table 2) [124]. 

### 4.1. DNA-PK Inhibitor

DNA-PKcs is a nuclear serine/threonine kinase and a critical protein that facilitates NHEJ [126]. Autophosphorylation of DNA-PKcs, which appears to be significant in NHEJ, causes a conformational shift that allows for end-processing enzymes to reach the ends of the DSBs [127]. DNA-PKcs work in conjunction with ATR and ATM to activate the phosphorylation of proteins involved in DNA damage checkpoints. Small compounds that target the AP-binding site of the kinase domain are the most effective methods of inhibiting DNA-PK to date [128]. The inhibition of DNA-PKcs would impede the kinase ability of DNA-PKcs and reduce the phosphorylation of cGAS. In addition, the use of DNA-PKcs inhibitors can sensitize cells to damaging agents [9,129]. 

Wortmannin has been used to inhibit DNA-PKcs. The clinical use of this substance is restricted by its lack of specificity, low solubility in aqueous solutions, and in vivo toxicity [130]. The plant flavonoid quercetin has a morpholine derivative, LY294002, which is also a widely used non-specific DNA-PK inhibitor. SCR7 is a small molecule that inhibits NHEJ [131]. NHEJ is eliminated because Ligase IV’s DNA-binding domain (DBD) is selectively bound by SCR7, preventing Ligase IV from attaching to the damaged chromatin. NHEJ is frequently overexpressed in various malignancies, which aids in resistance to several chemotherapeutic and radiation treatment methods. The other DNA-PK inhibitor, M3814, can be used to treat some types of cancer, alone or combined with other therapies, such as radiotherapy. For instance, M3814 can reduce tumor growth in combination with radiotherapy and the drug avelumab [129].

### 4.2. Poly ADP-Ribose Polymerase Inhibitor

A class of proteins called poly ADP-ribose polymerase (PARP) is involved in several biological processes such as DNA repair, genomic stability, and cell death [132]. Furthermore, BER pathways and SSB repair depend on PARP [133]. So far, many PARP families have been identified, and PARP-1 and PARP-2 proteins are essential for cell survival. PARP-1 and PARP-2 use NAD+ as a substrate to perform PARylation and release nicotinamide. These modifications regulate the conformation, stability, and activity of target proteins. Normal cells are repaired when damaged through HR, but cancer is fatal when PARP-1 is suppressed because the important proteins for HR, BRCA1 and BRCA2, are broken. PARP inhibitors engage in DNA repair and inhibit the ribosomes needed when cancer cells proliferate. DDX21 is required to produce ribosomes, while PARP-1 is necessary for DDX21 function. Therefore, PARP inhibitors can suppress cancer progression. Therefore, numerous treatments for breast, ovarian, prostate, and colon cancer are either undergoing clinical studies or are already being utilized in certain cases. Olaparib, lucaparib, niraparib, talazoparib, and celiparib are currently being used for cancer treatment [22]. The PARP inhibitor, in combination with immunotherapy, can be used to target the immune system to treat some types of cancer, such as ovarian, lung, gastrointestinal, and prostate cancers. For example, in ovarian cancer, the use of the PARP inhibitor can decrease the overall response rate (ORR) in patients (ORR range: 45–63%; range of the disease control rate (DCR) from the control sample: 73–81%) [129].

### 4.3. Ataxia Telangiectasia Mutated Inhibitor 

ATM plays a critical role in the HR repair system of DSB. ATM also controls cell cycle progression, transcriptional regulation, chromatin remodeling, and apoptosis. Various cofactors of ATM have been identified, including the MRN complex, TIP60, ING3, ATMIN, and WIP1. Inhibiting ATM factors sensitizes cells to IR and induces DSB. 

Numerous ATM inhibitors are currently being researched for cancer treatment. KU-55933 (2-morphin-4-yl-6-thianthren-1 yl-pyran-4-one) is an ATM inhibitor. Cancer cells exposed to KU-55933 were sensitized to the cytotoxic effects of IR and chemotherapeutic agents that induced DNA DSB, such as camptothecin, doxorubicin, and etoposide. Therefore, inhibition of ATM proteins is an alternative approach to suppressing tumor growth; in addition, compared to other DDR-targeted agents such as PARP inhibitors, the study of ATM inhibitors is still in the early stages [124]. Therefore, inhibition of ATM proteins is an alternative approach to suppressing tumor growth [124,134]. 

KU-59403, an upgraded version of KU-55933, can also be used for cancer treatment. This inhibitor has higher potential, solubility, and bioavailability than KU-55933. Though KU-59403 alone has no effect on tumor growth, it increases the anti-tumor effects of other inhibitors, such as topoisomerase inhibitors, when combined with them [103].

### 4.4. Ataxia Telangiectasia and Rad3-Related Inhibitor

ATR can sense stalled replication forks and induce various responses to DNA replication stress, which is important for maintaining the genomic integrity of cells. ATR also plays a role in the HR repair system in the presence of DSB along with interstand and cross-link repair systems. ATR is important for cell survival, particularly in the context of ATM mutations, making ATR a prospective target for cancer treatment [124,135,136]. Inhibiting ATR activity elevates the sensitivity of cancer cells to genotoxic agents and/or induces apoptosis. In addition, partial inhibition of ATR, resulting in cell stress, can cause aging in mice models [103].

NU6027, an ATR inhibitor, can increase the sensitivity of certain types of cancer, such as breast cancer, to irradiation and other cancer therapies [103]. However, the development of cancer therapy targeting the ATR signaling cascade was initially focused on CHK1 inhibitors rather than the ATR kinase itself. This may be due to the difficulty of obtaining the pure active form of the kinase protein BAY 1895344, which is a highly compelling and selective oral ATR inhibitor [135,136].

### 4.5. Checkpoint Kinase 1 Inhibitor

CHK1 plays an important role in DNA damage response and DNA damage repair. The phosphorylation of CHK1 by ATR mediates the repair process, and CHK1 delays the process of cell cycle progression, allowing cells to be repaired. Therefore, CHK1 acts as a cell-cycle checkpoint which can improve the survival rates of cells and increase the resistance of cancer cells to therapy [124,137].

Therefore, the regulation of CHK1 is used as an anticancer target in cancer therapy. Inhibition of CHK1 can result in cancer cell death by preventing the restart of stalled replication forks. Previous studies have identified numerous CHK1 inhibitors. The inhibition of CHK1 could increase the susceptibility of cancer cells to drugs, thus inducing replication stress in cancer cells. Some clinical studies have shown that CHK1 inhibitors can act as single agents to inhibit cancer cells and can work with other drugs or therapies to inhibit tumor growth [124,137,138,139,140].

There are two generations of CHK1 inhibitors. When combined with a cytotoxic agent, cancer cells showed sensitivity to the first generation of CHK1 inhibitors, but studies on this kind of inhibitor were limited due to its high toxicity. Second-generation CHK1 inhibitors have shown improvement compared to the first generation. LY2880070 and SRA737 are some of the CHK1 inhibitors that are currently under study. These drugs are being used in combination with other damaging agents, therapies, and antimetabolites [9]. 

## 5. Challenges and Perspective

Targeting the DNA repair pathway is an efficient method for cancer therapy. However, all DDR-related inhibitors under pre-clinical or clinical trials target enzymes, such as kinases (ATM, ATR, and DNA-PK) or PARP. Most regulators involved in the DDR pathway are scaffold proteins that are important for signal transduction, but without any enzyme activity, which complicates the design of small-molecule inhibitors for targeting these proteins. Therefore, the development of alternative strategies to target these untargetable scaffold proteins will broaden our options for cancer therapy. PROteolysis-TArgeting Chimeras (PROTACs) are a powerful class of compounds that selectively degrade proteins of interest through the cellular ubiquitination system. Recently, PROTACS targeting C-MYC, BET, androgen receptors, and BRD7 have effectively killed cancer cells [141,142]. Therefore, targeting scaffold proteins in the DDR pathway with PROTACs may also be a feasible method for cancer therapy. In addition to PROTACs, CRISPR/CAS9-mediated gene editing has been tested for cancer therapy in preclinical and clinical trials [143]. Inactivation of key scaffold proteins in the DDR pathway with CRISPR/CAS9 may enhance the efficiency of cancer chemotherapy or radiotherapy. Generally, these new technologies will afford us additional cancer therapy options, but further evaluation is required before their clinical application.

## Figures and Tables

**Figure 1 ijms-24-04741-f001:**
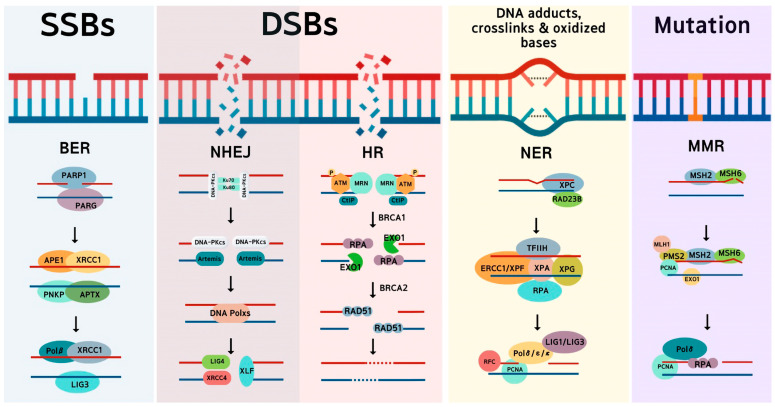
Types of DNA damage and their repair systems. Exogenous or endogenous insults cause strand breaks and base modification and mutation, and activate specific repair pathways corresponding to the types of DNA damage.

**Figure 2 ijms-24-04741-f002:**
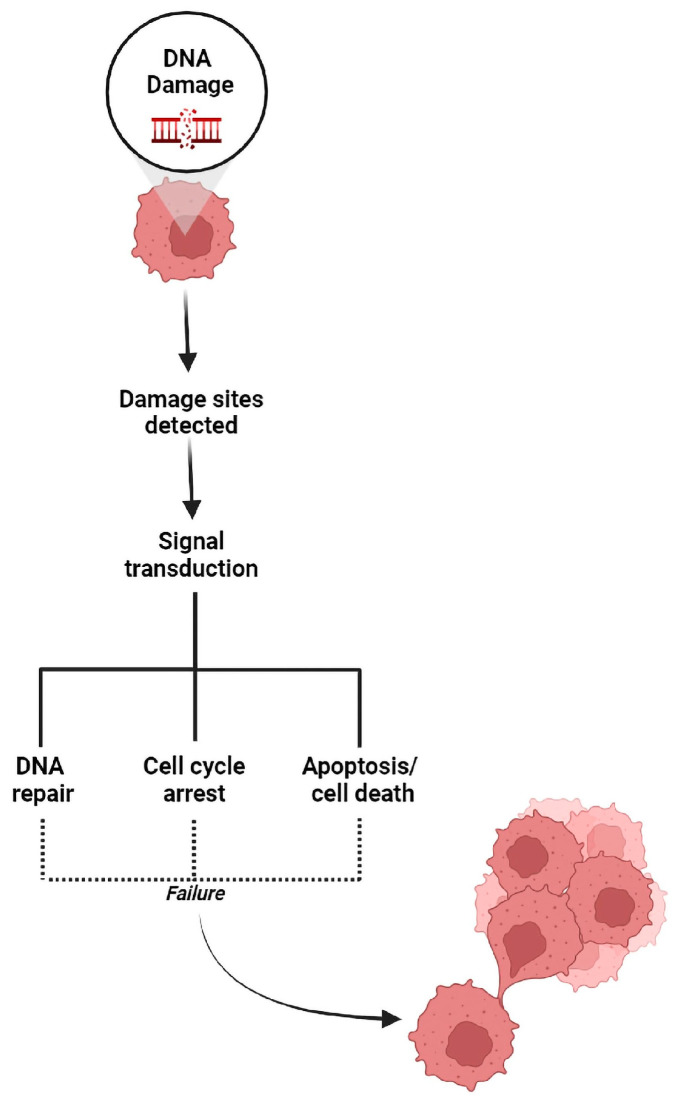
The DNA damage response is responsible for determining the cell’s future. DNA damage sensors recognize damage sites, transmit signals to related DNA damage repair systems, and determine whether the cell undergoes DNA damage repair, cell cycle arrest, or cell death. Any fault in this process could lead to genomic instability and cancer development.

**Table 2 ijms-24-04741-t002:** Cancer therapeutic treatment inhibitors and their targets, identified using the clinicaltrials.gov (4 January 2023) database.

Inhibitor Target	Clinical Trials. National Library of Medicine (NLM)	Clinical Phase	Disease	Intervention Title
DNA-PK	NCT02516813	Phase 1	Advanced Solid Tumors	Phase 1 Trial of MSC2490484A, an Inhibitor of a DNA-dependent Protein Kinase, in Combination with Radiotherapy
	NCT02316197	Phase 1	Advanced Solid Tumors Chronic Lymphocytic Leukemia	Clinical Phase I Study Investigating MSC2490484A, an Inhibitor of a DNA-dependent Protein Kinase, in Advanced Solid Tumors or Chronic Lymphocytic Leukemia
	NCT03770689	Phase 1 Phase 2	Locally Advanced Rectal Cancer	Study of Peposertib in Combination with Capecitabine and RT in Rectal Cancer
	NCT03907969	Phase 1 Phase 2	Advanced Malignancies	Clinical Trial to Evaluate AZD7648 Alone and in Combination with Other Anti-cancer Agents in Patients with Advanced Cancers
	NCT03724890	Phase 1	Oncology Solid Tumors	Study of Avelumab-M3814 Combinations
	NCT05002140	Phase 1	Metastasis Locally Advanced Solid Tumor Recurrent Cancer	Trial of XRD-0394, a Kinase Inhibitor, in Combination with Palliative Radiotherapy in Advanced Cancer Patients
PARP	NCT01844986	Phase 3	Newly Diagnosed Advanced Ovarian Cancer FIGO Stage III-IV BRCA Mutation Complete Response Partial Response First Line Platinum Chemotherapy	Olaparib Maintenance Monotherapy in Patients with BRCA Mutated Ovarian Cancer Following First Line Platinum Based Chemotherapy (SOLO-1)
	NCT02282020	Phase 3	Relapsed Ovarian Cancer, BRCA Mutation, Platinum Sensitivity	Olaparib Treatment in Relapsed Germline Breast Cancer Susceptibility Gene (BRCA) Mutated Ovarian Cancer Patients Who Have Progressed at Least 6 Months After Last Platinum Treatment and Have Received at Least 2 Prior Platinum Treatments (SOLO3)
	NCT02446704	Phase 1 Phase 2	Small Cell Lung Cancer	Study of Olaparib and Temozolomide in Patients with Recurrent Small Cell Lung Cancer Following Failure of Prior Chemotherapy
	NCT02789332	Phase 2	Breast Cancer Triple Negative Breast Neoplasms HRpos Breast Neoplasms BRCA 1/2 and/or HRD	Assessing the Efficacy of Paclitaxel and Olaparib in Comparison to Paclitaxel/Carboplatin, Followed by Epirubicin/Cyclophosphamide as Neoadjuvant Chemotherapy in Patients with HER2-negative Early Breast Cancer and Homologous Recombination Deficiency (GeparOla)
	NCT01989546	Phase 1 Phase 2	Advanced Ovarian Cancer Primary Peritoneal Cancer Advanced Breast Cancer Advanced Solid Tumors	Pilot Trial of BMN 673, an Oral PARP Inhibitor, in Patients with Advanced Solid Tumors and Deleterious BRCA Mutations
	NCT00494442	Phase 2	Ovarian Neoplasm	Study to Assess the Efficacy and Safety of a PARP Inhibitor for the Treatment of BRCA-positive Advanced Ovarian Cancer (ICEBERG 2)
	NCT01945775	Phase 3	Breast Neoplasms BRCA 1 Gene Mutation BRCA 2 Gene Mutation	A Study Evaluating Talazoparib (BMN 673), a PARP Inhibitor, in Advanced and/or Metastatic Breast Cancer Patients with BRCA Mutation (EMBRACA Study) (EMBRACA)
	NCT04174716	Phase 1 Phase 2	Solid Tumors Homologous Recombination Repair Gene Mutation Homologous Recombination Deficiency	Basket Trial of IDX-1197, a PARP Inhibitor, in Patients with HRR Mutated Solid Tumors (VASTUS) (VASTUS)
	NCT04586335	Phase 1	Ovarian Cancer Breast Cancer Solid Tumor Prostate Cancer Endometrial Cancer	Study of CYH33 in Combination with Olaparib, an Oral PARP Inhibitor, in Patients with Advanced Solid Tumors
	NCT04972110	Phase 1 Phase 2	Advanced Solid Tumor, Adult	Study of RP-3500 with Niraparib or Olaparib in Advanced Solid Tumors (ATTACC)
ATM	NCT04882917	Phase 1	Advanced Solid Tumors	First-in-human Study of M4076 in Advanced Solid Tumors (DDRiver Solid Tumors 410)
	NCT03423628	Phase 1	Recurrent Glioblastoma Multiforme Primary Glioblastoma Multiforme Brain Neoplasms, Malignant Leptomeningeal Disease (LMD)	A Study to Assess the Safety and Tolerability of AZD1390 Given with Radiation Therapy in Patients with Brain Cancer
	NCT02588105	Phase 1	Advanced Solid Tumors	Study to Assess the Safety and Preliminary Efficacy of AZD0156 at Increasing Doses Alone or in Combination with Other Anti-cancer Treatment in Patients with Advanced Cancer (AToM)
	NCT03225105	Phase 1	Solid Tumors	M3541 in Combination with Radiotherapy in Subjects with Solid Tumors
ATR	NCT03188965	Phase 1	Advanced Solid Tumor Non-Hodgkin’s Lymphoma Mantle Cell Lymphoma	First-in-human Study of ATR Inhibitor BAY1895344 in Patients with Advanced Solid Tumors and Lymphomas
	NCT04267939	Phase 1	Advanced Solid Tumors (Excluding Prostate Cancer) Ovarian Cancer	ATR Inhibitor Elimusertib (BAY1895344) Plus Niraparib First phase b Study in Advanced Solid Tumors and Ovarian Cancer
	NCT05338346	Phase 1	Advanced Solid Tumors Hematological Malignancies	A Study of ATG-018 (ATR Inhibitor) Treatment in Patients with Advanced Solid Tumors and Hematological Malignancies (ATRIUM)
	NCT04065269	Phase 2	Gynecological Cancers	ATR Inhibitor in Combination with Olaparib in Gynecological Cancers with ARId1A Loss or no Loss (ATARI)
	NCT05071209	Phase 1 Phase 2	Recurrent Alveolar Rhabdomyosarcoma Recurrent Ewing Sarcoma Recurrent Lymphoma Recurrent Malignant Solid Neoplasm Refractory Alveolar Rhabdomyosarcoma Refractory Ewing Sarcoma Refractory Lymphoma Refractory Malignant Solid Neoplasm	Elimusertib for the Treatment of Relapsed or Refractory Solid Tumors
	NCT04972110	Phase 1 Phase 2	Advanced Solid Tumor, Adult	Study of RP-3500 with Niraparib or Olaparib in Advanced Solid Tumors (ATTACC)
	NCT04497116	Phase 1 Phase 2	Advanced Solid Tumor	Study of RP-3500 in Advanced Solid Tumors
	NCT05269316	Phase 1	Solid Tumor Advanced Solid Tumor	Study to Evaluate IMP9064 as a Monotherapy or in Combination in Patients with Advanced Solid Tumors
	NCT03787680	Phase 2	Prostate Cancer	Targeting Resistant Prostate Cancer with ATR and PARP Inhibition (TRAP Trial)
CHK1	NCT02797964	Phase 1 Phase 2	Advanced Solid Tumors or Non-Hodgkin’s Lymphoma (NHL)	A First phase/2 Trial of SRA737 in Subjects with Advanced Cancer
	NCT02797977	Phase 1 Phase 2	Advanced Solid Tumors	A First phase/2 Trial of SRA737 in Combination with Gemcitabine Plus Cisplatin or Gemcitabine Alone in Subjects with Advanced Cancer
	NCT03057145	Phase 1	Solid Tumor	Combination Study of Prexasertib and Olaparib in Patients with Advanced Solid Tumors
	NCT00045513	Phase 1 Phase 2	Leukemia Lymphoma	Combination Chemotherapy in Treating Patients with Chronic Lymphocytic Leukemia or Lymphocytic Lymphoma
	NCT00700336	Phase 1 Phase 2	Malignant Pleural Mesothelioma Solid Tumors	Study of CBP501 + Pemetrexed + Cisplatin on MPM (Phase I/II)
	NCT00415636	Phase 1	Cancer	Safety and Tolerability of IC83/LY2603618 Administered After Pemetrexed 500 mg/m^2^ Every 21 Days in Patients with Cancer
	NCT00839332	Phase 1 Phase 2	Pancreatic Neoplasms	A Study for Participants with Pancreatic Cancer
	NCT00988858	Phase 2	Non-Small Cell Lung Cancer	A Study of Advanced or Metastatic Non-Small Cell Lung Cancer
	NCT01296568	Phase 1	Advanced Cancer	C14 Study in Oncology Patients with Advanced and/or Metastatic Solid Tumors
	NCT01139775	Phase 1 Phase 2	Non-Small Cell Lung Cancer	A Study in Non-Small Cell Lung Cancer
	NCT01341457	Phase 1	Solid Tumors	A Study of LY2603618 in Combination with Gemcitabine in Participants with Solid Tumors
	NCT01358968	Phase 1	Cancer	A Drug Interaction Study to Assess the Effect of LY2603618 on the Metabolic Pathway of Desipramine
	NCT00779584	Phase 1	Hodgkin Disease Lymphoma, Non-Hodgkin Neoplasms	A Dose-Escalation Study of MK-8776 (SCH 900776) with and without Gemcitabine in Participants with Solid Tumors or Lymphoma (MK-8776-002/P05248)
	NCT01115790	Phase 1	Advanced Cancer Squamous Cell Carcinoma Carcinoma, Squamous Cell of Head and Neck Lung Squamous Cell Carcinoma Stage IV Anal Squamous Cell Carcinoma Carcinoma, Non-Small-Cell Lung	A First Phase Study in Participants with Advanced Cancer
	NCT04095221	Phase 1 Phase 2	Desmoplastic Small Round Cell Tumor Rhabdomyosarcoma	A Study of the Drugs Prexasertib, Irinotecan, and Temozolomide in People with Desmoplastic Small Round Cell Tumor and Rhabdomyosarcoma
	NCT03495323	Phase 1	Cancer	A Study of Prexasertib (LY2606368), CHK1 Inhibitor, and LY3300054, PD-L1 Inhibitor, in Patients with Advanced Solid Tumors
	NCT03057145	Phase 1	Solid Tumor	Combination Study of Prexasertib and Olaparib in Patients with Advanced Solid Tumors
	NCT02873975	Phase 2	Advanced Cancers	A Study of LY2606368 (Prexasertib) in Patients with Solid Tumors with Replicative Stress or Homologous Repair Deficiency
	NCT02808650	Phase 1	Childhood Solid Neoplasm Recurrent Malignant Solid Neoplasm Recurrent Primary Central Nervous System Neoplasm Refractory Malignant Solid Neoplasm Refractory Primary Central Nervous System Neoplasm	Prexasertib in Treating Pediatric Patients with Recurrent or Refractory Solid Tumors
	NCT02514603	Phase 1	Neoplasm	A Study of Prexasertib (LY2606368) in Japanese Participants with Advanced Cancers

## Data Availability

Not applicable.

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
