# Peer review of "DNA Damage and Its Role in Cancer Therapeutics"

_ijms, 2023, doi:10.3390/ijms24054741_

Round 1
Reviewer 1 Report
This manuscript includes both the advantages and disadvantages of DNA damage to describe the general pathway of DNA repair in cancer cells and potential proteins that may be targets for cancer treatment. Detailed and comprehensive introduction to the main ways of DNA repair. The overall structure and logic of the article are relatively clear, which better demonstrates DNA damage and its important role in cancer treatment. However, there are still many aspects to be improved, which are reflected in the following aspects.
Comments needed to be focused on:
1. In the part of ' Main DNA repair pathways in cancer cells ', the author classified the DNA repair pathways in cancer cells in detail, but the author 's summary was lacking, and the relationship between cancer and DNA repair pathways was not discussed in depth. Only the DNA repair pathway is introduced, which is not linked to cancer.
2. Figure 1 shows the picture of DNA damage types, but it lacks the picture of DNA damage repair pathway. It is suggested that the author add the corresponding repair path pictures to make the content more detailed and complete.
3. The logic of the third part is chaotic. Although the article introduces many mutations of genes that promote and inhibit cancer, it seems to deviate somewhat from the theme, and the content description is loose and confusing, which is not easy for readers to understand.
4. The fourth part should be the core content, but the author is not closely combined with the clinical aspects and the reference of the literature is also lacking, and the content introduction is relatively superficial.
5. Line 99 - what exactly does XLF mean? It is suggested to add one or two sentences to introduce it.
6. Line 203-205, In my opinion, it is inappropriate to put this summary in this place. It is the impact of cancer on DNA damage, rather than the impact of DNA damage on cancer as the author's theme explains.
7. Line 214-222,The author should appropriately add examples between DNA repair and cancer cell progression, such as [55, 56]. This will be more conducive to the proof of the third part.
8. The image resolution is poor, you can add figure legends under the name, so that readers can understand.
9. In Figure 3, why there are two ‘Primary repair proteins’ in Figure 3?
10. Line 243 - It is suggested to add a sentence at the beginning to clarify the relationship between synthetic lethality and DDR, so that readers can understand the intention of this paragraph.
11. Line 249 - What do ‘These pathways’ concretely mean ?
Therefore, I suggest the manuscript can be published after major revision.
Reviewer 2 Report
The review is well structured and covers the main aspects of DNA damage and repair mechanisms. However, I have the following suggestions:
- The article needs editing for English
- Figures to show the protein complexes involved and their roles in HDR, NHEJ, MMEJ, BER, NER, MMR - at present this information is in the text but it is dense reading
- BRCA genes can be inactivated by deletion and promoter methylation - this is not reflected in Table 1 or in the text
Round 2
Reviewer 1 Report
In the latest manuscript, you revised the article substantially in response to the questions that I asked for the first time. I think the revised manuscript is of a much improved quality than before and has met the publication criteria of this journal. Therefore I recommend agreeing to accept the manuscript.